# Dataset Size Recovery from Fine-Tuned Weights

## Abstract

Model inversion and membership inference attacks aim to reconstruct and verify the data on which a model was trained. However, these methods cannot guarantee to find all training samples, as they do not know the training set size. In this paper, we introduce a new task: *dataset size recovery*, which seeks to identify the number of samples a given model was fine-tuned on. Our core finding is that both the norm and the spectrum of the fine-tuning weight matrices are closely linked to the fine-tuning dataset size. Leveraging this insight, we propose DSiRe, an algorithm that accepts fine-tuned model weights, extract their spectral features, and then employs a nearest neighbor classifier on top, to predict the dataset size. Although it is training-free, simple, and very easy to implement, DSiRe is broadly applicable across various fine-tuning paradigms and modalities (e.g., DSiRe can predict the number of fine-tuning images with a mean absolute error of 0.36 images). To this end, we develop and release *LoRA-WiSE*, a new benchmark consisting of over $25k$ weight snapshots from more than $2k$ diverse LoRA fine-tuned models.

## 1 Introduction

Data is the top factor for the success of machine learning models. Model inversion (Fredrikson et al., 2015; Yang et al., 2019; Haim et al., 2022) and membership inference attacks (Carlini et al., 2022; Shafran et al., 2021; Jagielski et al., 2024) aim to reconstruct and verify the training data of a model, using its weights (Haim et al., 2022; Duan et al., 2023; Nguyen et al., 2023). While these methods may discover *some* of the training data, they are not guaranteed to recover *all* training samples. One fundamental limitation that prevents them from discovering the entire training data is that they do not have a halting condition, as they do not know the size of the training set (Haim et al., 2022). E.g., in membership inference, the attacker sequentially tests samples for membership in the training set, but without knowing the dataset size, it's difficult to establish the halting condition or how many samples should be classified as "in" the training set. Knowing the training set size provides a crucial halting condition and provides a principled way to set thresholds. Discovering the size of a training dataset given the model weights is important, even without explicit reconstruction of the images themselves. Understanding the number of images used to train or fine-tune models is also of great interest to researchers, who wish to understand the number of data needed to replicate a model. We therefore propose a new task: *Dataset Size Recovery*, which aims to recover the number of training samples a given model was fine-tuned on.

To tackle this challenge, we begin by analyzing the relationship between dataset sizes and the corresponding weights of fine-tuned models. As demonstrated in Fig. 2a, the Frobenius norm of the fine-tuned weights is highly correlative with the corresponding dataset size. Specifically, the norm decreases as the dataset size increases. However, since representing the weights matrix using only the Frobenius norm has limited expressivity (as it's only a single scalar), we proceed to extract singular values from the weights, providing more expressive features. In Fig. 2b we can see that this approach shares the same phenomenon that we previously described.

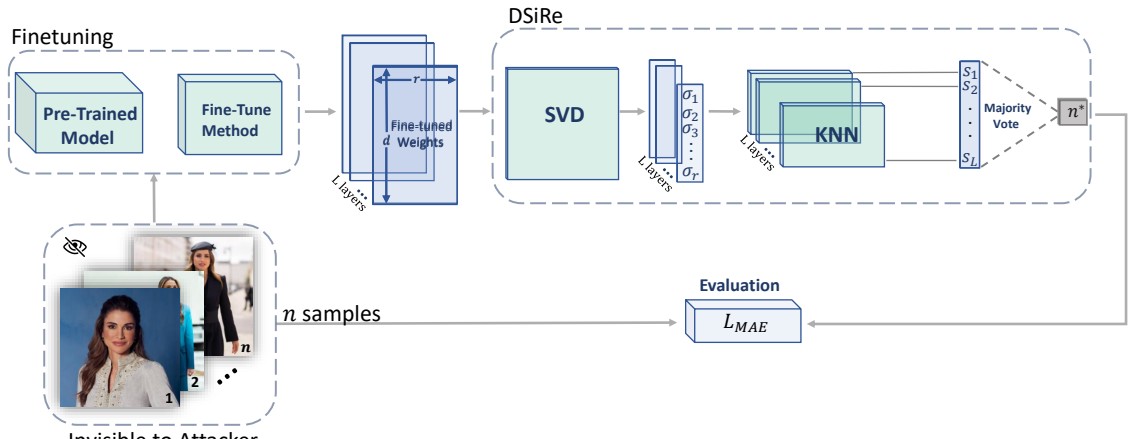

Figure 1: **DSiRe:** We introduce the task of *dataset size recovery*, which aims to recover the dataset size used to fine-tune a model based on its weights. extracts the singular values of each weight matrix and treats them as features. These features are then used to train a set of layer-specific nearest-neighbor classifiers which predict the dataset size.

We therefore introduce *DSiRe* (**D**ataset **Si**ze **Re**covery), an algorithm for recovering the dataset size across a wide range of fine-tuned models. Given a fine-tuned model, *DSiRe* represents it by extracting the singular values from each of its layers' weights matrices. This representation yields a feature space for exploring various classification techniques. In particular, the best version of *DSiRe* uses the very simple baseline of nearest neighbors classification to label each model layer independently. The model-level prediction is then determined by the majority vote of its layers (See Fig. 1 for an overview). With merely weights information, it is effective for models fine-tuned on different modalities (e.g., vision and language), different tasks (e.g., generative and discriminative), and with various architectures (e.g., transformers, CNNs, and MLPs). Moreover, it is not limited to a single fine-tuning paradigm, working well whether the model is fully fine-tuned or partially adapted (e.g. LoRA (Hu et al., 2021), DoRA (Liu et al., 2024)).

To evaluate DSiRe and encourage future research, we introduce *LoRA-WiSE*, a new, large-scale, and diverse dataset. *LoRA-WiSE* comprises over $25k$ weights checkpoints drawn from more than $2k$ independent LoRA models, spanning different dataset sizes, backbones, ranks, and personalization sets. On LoRA-WiSE, DSiRe recovers the dataset sizes with a Mean Absolute Error (MAE) of $0.36$, demonstrating that our method is highly effective in realistic settings.

To summarize, our main contributions are:

1. Introducing the task of dataset size recovery.
2. Presenting a method for recovering dataset size for fine-tuned models.
3. Releasing LoRA-WiSE, the first dataset size recovery evaluation suite.

## 2 RELATED WORK

**Predictions from Neural Network Weights.** Predicting neural network attributes directly from their weights is a relatively new and challenging area of research. Eilertsen et al. (2020) and Unterthiner et al. (2020)

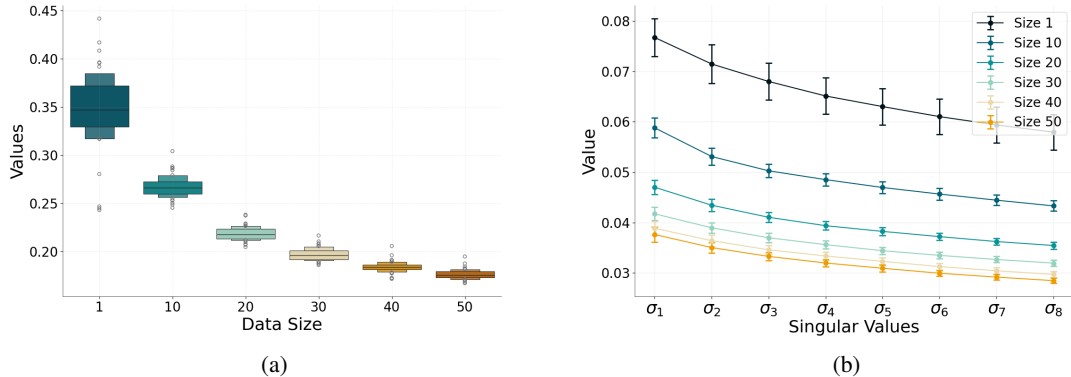

(a)              (b)

Figure 2: *Norm and Spectrum of Fine-Tuning Weights vs. Dataset Size.* Analysis of 300 Stable Diffusion 1.5 models fine-tuned on datasets of sizes from $[1, 10, 20, 30, 40, 50]$. (a) Frobenius norm range per dataset size (b) Singular values per dataset size. There is a clear negative correlation between weight/spectrum magnitudes and the size of the fine-tuning dataset.

pioneered this approach, predicting training hyperparameters and generalization capabilities, respectively. More recently, (Zhou et al., 2024) proposed a permutation-invariant neural network weight encoder for performance prediction. These works primarily focus on predicting properties of the entire network or its performance. However, most of these methods have been limited to small networks (e.g., small MLPs or CNNs with 3-5 layers) and have not been scaled to foundation models. Our work introduces a new task: recovering the dataset size used for fine-tuning large-scale models. Our proposed method, DSiRe, leverages the spectrum of the weights, demonstrating the potential of weight-space analysis for foundation models and opening new avenues for understanding fine-tuned models.

**Model Fine-Tuning.** Model fine-tuning (Zhang et al., 2023a; Zhai et al., 2022; Avrahami et al., 2023b) adapts a model for a downstream task and is considered a cornerstone in machine learning. The emergence of foundation models (Radford et al., 2021; Touvron et al., 2023; Brown et al., 2020; Rombach et al., 2022) has made standard fine-tuning costly and unattainable without substantial resources. Parameter-Efficient Fine-Tuning (PEFT) methods were then proposed (Hu et al., 2021; Dettmers et al., 2023; Houlsby et al., 2019; Li & Liang, 2021; Lester et al., 2021; Liu et al., 2023; He et al., 2021; Liu et al., 2022; Jia et al., 2022; Zhang et al., 2023b; Wang et al., 2023a; Hyeon-Woo et al., 2021), offering various ways to fine-tune models with fewer optimized parameters. Among these methods, LoRA (Hu et al., 2021) stands out, proposing to train additive low-rank weight matrices while keeping the pre-trained weights frozen. LoRA was found to be very effective across several modalities (Wang et al., 2023b; Ye et al., 2023; Avrahami et al., 2023a). Recently, (Horwitz et al., 2024) identified a security issue with LoRA, demonstrating that multiple LoRAs can be used to recover the original pre-trained weights. In this paper, we uncover a new use case of LoRA fine-tuning, specifically focusing on the recovery of the dataset size from text-to-image models fine-tuned via LoRA.

**Membership Inference & Model Inversion Attacks.** Two privacy vulnerabilities found in machine learning models are Membership Inference Attack (MIA) (Salem et al., 2018; Carlini et al., 2022; Hu et al., 2022; Shafran et al., 2021; Jagielski et al., 2024) and Model Inversion (Fredrikson et al., 2015; Yang et al., 2019; He et al., 2019; Yin et al., 2020; Haim et al., 2022). First presented by (Shokri et al., 2017), MIAs aim to verify whether a certain image was in the training dataset of a given model. Typically, MIAs assumes that training samples are over-fitted proposing various membership criteria; either by looking for lower loss values (Sablayrolles et al., 2019; Yeom et al., 2018) or some other metrics (Watson et al., 2021; Carlini et al., 2022). In generative models, MIAs have been extensively researched as well (Hilprecht et al., 2019; Hayes et al.,

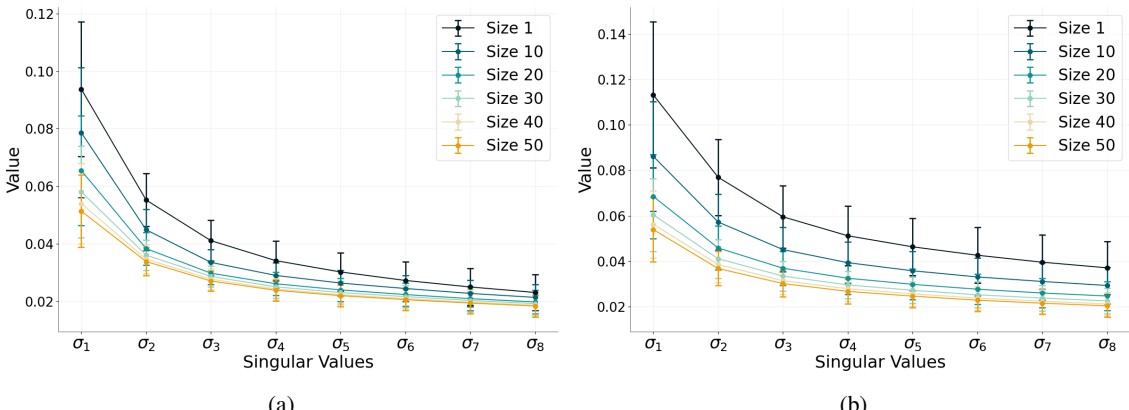

(a)                                                              (b)

Figure 3: ***Spectrum Ranges of 2 Different Layers.*** Singular values distribution of two layers on opposite sides of Stable Diffusion 1.5 UNet, fine-tuned on datasets of sizes $[1, 10, 20, 30, 40, 50]$. (a) First down block (b) Last upper block. The last upper block shows greater separation of singular values compared to the first down block, highlighting that not all layers are born equally for dataset size recovery.

2017; Chen et al., 2020), including recent attacks against diffusion models (Matsumoto et al., 2023; Hu & Pang, 2023).

*Model inversion* is a similar attack, in a data-free setting. Introduced by (Fredrikson et al., 2015), model inversion methods wish to generate training samples from scratch, instead of asking whether a known specific image was in the training set. Model inversion is also used for settings where data is unavailable, e.g., data-free quantization (Choi et al., 2021; Xu et al., 2020; Li et al., 2023) and data-free distillation (Lopes et al., 2017; Zhu et al., 2021; Zhang et al., 2022; Fang et al., 2022; Shao et al., 2023).

(Haim et al., 2022) emphasized the importance of recovering the training set size for model inversion applications. When this size is unknown, it prevents model inversion attacks from reconstructing the entire dataset a model was trained on, as it is unclear how many samples are sufficient. Our work specifically addresses this issue by uncovering a new vulnerability in fine-tuned models, which enables us to infer the size of the dataset used for fine-tuning.

## 3 MOTIVATION

**Frobenius Norm Analysis.** Our hypothesis is that the difference between pre-fine-tuning and post-fine-tuning weights, denoted as $\Delta W$, encodes valuable information about the size of the fine-tuning dataset. To investigate this, we first consider a simple statistic of each fine-tuning matrix; its Frobenius norm, $s_F$, defined as:

$$s_F = \sum_{ij} |\Delta W_{ij}|^2 \tag{1}$$

The norm of a weight matrix is known to correlate with the expressivity of the network. For example, weight decay, a common regularization technique, effectively constrains this norm. To analyze the correlation between the Frobenius norm and the fine-tuning dataset size, We conducted an experiment using Stable Diffusion (SD) 1.5. We fine-tuned the model on 50 micro-datasets of sizes $[1, 10, 20, 30, 40, 50]$ images, while keeping all other hyper-parameters fixed. Fig. 2a shows the range of values for the Frobenius norm statistic $s_F$ across different dataset sizes. The results clearly demonstrate a negative correlation between $s_F$

and the dataset size. We motivate this correlation by over-fitting, i.e., models tend to overfit faster on smaller dataset sizes, leading to larger values of $s_F$.

**Analyzing the Singular Value Spectrum.** To gain a deeper understanding, we extended our analysis to the singular value spectrum of the fine-tuning matrix. Fig. 2b visualize the $m^{th}$ singular value (denoted as $\sigma_m$) for different dataset sizes. We note there is a better separation between different dataset sizes for the largest singular values. This suggests that the spectrum is more discriminative than the scalar Frobenius norm. Overall, both $s_F$ and the spectrum indicate larger values for smaller dataset sizes.

**Layer-specific Analysis.** Finally, we analyzed how discriminative different layers are for predicting fine-tuning dataset size. We plot the spectra of layers in two distinct blocks of the UNet architecture: the first down block and the last up block. For each block, we calculated the mean and standard deviation of singular values across all layers and present the results in Fig. 3. We can see that the up layer is more discriminative than the down one, perhaps suggesting that the UNet decoder is more prone to over-fitting than the encoder. However, it's important to note that our experiments showed no single layer is universally discriminative across all models. Therefore, we conclude that combining results from all layers yields the most robust prediction of dataset size.

## 4 METHOD

### 4.1 TASK DEFINITION: DATASET SIZE RECOVERY

We introduce the task of Dataset Size Recovery for fine-tuned dataset, a new attack vector against fine-tuned models. Formally, given the fine-tuning weights of all layers of a model denoted as $\Delta \mathcal{W} = [\Delta W_1, \Delta W_2, ...\Delta W_L]$, our task is to recover the number of images $n$ that the model was fine-tuned on. More formally, we wish to find a function $f$, such that:

$$n = f(\Delta \mathcal{W}) \tag{2}$$

The effectiveness of this attack was measured by the MAE between $f(\Delta \mathcal{W})$ and $n$ across a set of models.

### 4.2 DSiRe

We propose DSiRe (**D**ataset **Si**ze **Re**covery), a supervised method for recovering dataset size from fine-tuned weights. Our approach first constructs a training dataset by fine-tuning multiple models on concept personalization sets across a range of dataset sizes. It then trains a predictor function $f$ that operates on a set of fine-tuning weights of each model and outputs the predicted dataset size $n$. At test time, it generalizes to unseen models trained with different concepts. The method can be seen in Fig. 1

**Training set synthesis.** We first synthesize a training set by fine-tuning our model on each of $N_{train}$ datasets, each containing a set of training images. The datasets span a range of sizes; in this paper, we tested the ranges $1-6$, $1-50$, and $1-1000$. The result is a set of $N_{train}$ models $\Delta \mathcal{W}_m$, each with a corresponding label of the dataset size $n_m$.

**Constructing DSiRE.** Given the set of $N_{train}$ labeled models, we wish to train a predictor that maps the fine-tuning weights $\mathcal{W}_m$ to dataset size $n_m$. Motivated by the results of our analysis (see Sec. 3), we represent a given model as the set of spectra of all of its $L$ layers:

$$\Psi = [\Sigma_1, \Sigma_2, \ldots, \Sigma_L]$$

where $\Sigma_i$ denotes the singular values of the weight matrix of the fine-tuning layer$_i$. During inference, given a new model, we label each layer$_i$ using the label of the nearest i'th layer in the train set. Then, we get the overall prediction by ensembling the results of the model layers using the majority vote. In practice, we tested many different methods for labeling each layer in $\Psi$ and ablate them in Fig. 4. Overall, the simple Nearest Neighbor (NN) method performed the best.

## 5 EXPERIMENTS

### 5.1 EXPERIMENTAL SETUP

**Dataset.** Constructing a dataset of real-world foundation models is challenging due to computational and storage limitations. Therefore, we utilize LoRA fine-tuning, which is emerging as the most popular fine-tuning paradigm for foundation models, and propose the LoRA Weight Size Evaluation (LoRA-WiSE) benchmark, a comprehensive dataset designed to evaluate dataset size recovery methods for generative models. LoRA-WiSE comprises 2,350 Stable Diffusion models (versions 1.5 and 2) fine-tuned using LoRA across various dataset sizes, ranging from 1 to 1000 images. The benchmark includes multiple data ranges, LoRA ranks, and backbones to ensure diverse evaluation scenarios. Unless otherwise stated, we use Stable Diffusion (SD) 1.5 as the pre-trained backbone and a LoRA rank of 32.

**Settings.** *LoRA-WiSE* comprises of 3 data regimes: low ($\{1, 2, 3, 4, 5, 6\}$ samples), medium ($\{1, 10, 20, 30, 40, 50\}$ samples), and high ($\{1, 50, 100, 500, 1000\}$ sample). For each regime, we use 50 micro-datasets of different concepts (e.g., toys dataset, dogs dataset) to fine-tune SD on. Specifically, for each dataset size $s$, we sample $s$ images from each dataset, then fine-tune SD 1.5 on these resulted samples. This yields 300 fine-tuned models for the low and medium settings, and 250 models for the high setting. We then split each setting into train and test splits, with 15 models (fine-tuned on 15 micro-datasets) of each size to train, and the rest left for test. This results in (90, 210) split for the low and medium settings and (75, 175) split for the high setting. For a detailed description of the LoRA-WiSE benchmark and how these models are trained, see App. C.

In addition to LoRA-WiSE, we train a set of models to evaluate *full fine-tuning*. we focus on the difference between the original and fine-tuned weight matrices. Due to computational resources, we use the medium data regime, resulting in 90 models for training and 210 for testing.

To ensure robust evaluation and test generalizability across different data distributions, we repeat each experiment 10 times. In each time, we use subset sampling from models with varying object classes. We report the average and standard deviation across these experiments.

**Baseline.** We compare DSiRe to a baseline, denoted as Frobenius-NN, which predicts the dataset size using a nearest neighbor classifier on top of the Frobenius norms of the layers' LoRA weights. Similar to DSiRe, the Frobenius-NN is fitted separately to each layer, and then a majority vote rule is applied to select the prediction from all layer-wise predictions. The analysis in Sec 3 provides motivation for this baseline.

**Evaluation metrics.** As described in Sec. 4.1, our main evaluation metric is Mean Absolute Error (MAE). For completeness, we choose to report two complementary metrics as well: (i) Accuracy. (ii) Mean Absolute Percentage Error (MAPE). Since DSiRe predicts dataset sizes, simple accuracy does not adequately measure its effectiveness, e.g., predicting 4 when the true value is 5 is not as bad as predicting 1. We therefore provide MAPE scores as well, which compute the percentile from the ground truth that is equal to the absolute error.

### 5.2 RESULTS

**LoRA-WiSE.** We test DSiRe on a range of practical LoRA settings: We begin with a *low range* $1 - 6$ fine-tuning images, aiming to assess our method's performance on very small datasets. Results in Tab. 1 reveal that DSiRe outperforms Frobenius-NN by a small margin ($> 3\%$). This demonstrates DSiRe's ability to generalize well even on small, continuous ranges, making it suitable for scenarios with limited data availability.

Expanding our investigation to *mid-range* dataset sizes ($1 - 50$ images), which are common in artistic LoRA fine-tuning, we aim to evaluate DSiRe's performance on more typical use cases. Tab. 1 shows that DSiRe performs well with an MAE of $1.48$. In this data range, the Frobenius-NN baseline achieves comparable results to DSiRe across all metrics, demonstrating good performance. While the absolute MAE value is larger

Table 1: ***Performance Comparison of Dataset Size Recovery Methods on Full-Fine-Tuning and LoRA Paradigms.*** Performance of Frobenius-NN and DSiRe across different fine-tuning paradigms: full-fine-tuning (FFT) and LoRA, as well as different data ranges $(1-6, 1-50, 1-1000)$ for LoRA. This supports our analysis (see Sec. 3), which demonstrate that both singular values and Frobenius norm are indeed predictive of the dataset size. However, DSiRe outperfomrs the Frobenius-NN on all evaluation metrics.

| FT Paradigm | Data Range | Method | MAE $\downarrow$ | MAPE(%) $\downarrow$ | Acc(%) $\uparrow$ |
|---|---|---|---|---|---|
| Full-fine-tuning | 1-50 | Frobenius-NN | 1.91 $_{\pm 0.5}$ | 18.95 $_{\pm 3.1}$ | 82.84 $_{\pm 3.08}$ |
| | | DSiRe | **1.46** $_{\pm 0.24}$ | **5.99** $_{\pm 0.77}$ | **86.03** $_{\pm 2.05}$ |
| LoRA | 1-6 | Frobenius-NN | 0.43 $_{\pm 0.04}$ | 15.14 $_{\pm 2.12}$ | 65.29 $_{\pm 2.42}$ |
| | | DSiRe | **0.36** $_{\pm 0.04}$ | **11.36** $_{\pm 1.55}$ | **69.30** $_{\pm 3.83}$ |
| | 1-50 | Frobenius-NN | 1.56 $_{\pm 0.19}$ | 4.16 $_{\pm 0.75}$ | 85.33 $_{\pm 1.81}$ |
| | | DSiRe | **1.48** $_{\pm 0.21}$ | **3.97** $_{\pm 0.73}$ | **86.10** $_{\pm 1.99}$ |
| | 1-1000 | Frobenius-NN | 68.62 $_{\pm 5.53}$ | 9.25 $_{\pm 1.21}$ | 86.51 $_{\pm 1.12}$ |
| | | DSiRe | **41.77** $_{\pm 6.61}$ | **5.96** $_{\pm 1.46}$ | **91.90** $_{\pm 1.28}$ |

than in the low data range case, it is relatively small compared to the range of data sizes. The accuracy and mean absolute percentage error (MAPE) scores of both methods further support this observation. Fig. 6 shows another favorable property of our approach: its mistakes are usually near hits, i.e., large errors between ground truth and predicted labels are rare.

In larger data quantities, dataset size recovery could aid in better understanding data collection quantities needed for fine-tuning. Therefore, we conducted an additional experiment using models trained with *higher data ranges*, having $1, 50, 100, 500$ and $1000$ image samples per model (note that here we have 5 dataset size classes). Results, presented in Tab. 1, shows DSiRe is able to detect the dataset size with more than $90\%$ accuracy, and a MAPE score of only $6\%$. Additionally, in Fig. 7 we show the confusion matrix generated by DSiRe, where we see that most of the errors happen between adjacent classes.

**Continuous Range.** Motivated by the common occurrence of continuous dataset sizes in real-world scenarios, we aimed to evaluate DSiRe's performance in a regression setting. Our goal was to demonstrate that DSiRe can accurately predict dataset sizes from a continuous range, rather than just predefined discrete classes. We conducted an experiment using Stable Diffusion 1.5 as our base model. We created 20 datasets, each with a randomly sampled number of images from the range 1-40, and trained a LoRA on each dataset. This process was repeated 20 times, resulting in 400 LoRAs, each labeled with its specific training dataset size. For inference, we used DSiRE with k=2. Results in Tab. 2 show that DSiRe achieves an R² of 0.97. This demonstrates DSiRe's exceptional generalization to continuous ranges of dataset sizes, validating its practical utility in real-world scenarios where dataset sizes vary continuously.

Table 2: ***DSire Performance on Continuous Range*** This demonstrates DSiRe's exceptional generalization to continuous ranges of dataset sizes, validating its practical utility in real-world scenarios where dataset sizes vary continuously.

| Method | MAE $\downarrow$ | MAPE(%) $\downarrow$ | R²(%) $\uparrow$ |
|---|---|---|---|
| DSiRe | 1.02 $_{\pm 0.26}$ | 4.19 $_{\pm 2.29}$ | 0.97 $_{\pm 0.02}$ |

**Full fine-tuning.** We proceed to evaluate DSiRe on the full fine-tuning setting described in Sec. 5.1. The results in Tab. 1 show that both DSiRe and the Frobenius norm achieved good results, with DSiRe outperforming Frobenius-NN by a small margin. This is in line with our hypotheses from Sec. 3.

**Other Backbone.** The LoRA fine-tuning technique is commonly used by popular text-to-image models. A desirable aspect of our paradigm is being robust to model architecture. In this part, We test the robustness of DSiRe to the backbone model by evaluating it on Stable Diffusion 2.0. We note that these models do not share

Table 3: ***DSiRe's Versatility across Domains.*** Performance comparison between DSiRe and Frobenius-NN on diverse architectures and tasks: different backbone (SD 2), language models (GPT-2), discriminative (ResNet-50). Our method works well in all cases.

| Model | Data Range | Method | MAE ↓ | MAPE(%) ↓ | Acc(%) ↑ |
|-------|-----------|--------|-------|-----------|----------|
| SD 2.0 | [1, 10, 20, 30, 40, 50] | Frobenius-NN | 2.95 $_{\pm 0.28}$ | 11.99 $_{\pm 3.93}$ | 73.90 $_{\pm 2.21}$ |
| | | DSiRe | **2.30** $_{\pm 0.24}$ | **6.90** $_{\pm 0.82}$ | **79.95** $_{\pm 1.66}$ |
| GPT-2 | [1, 50, 250, 1000] | Frobenius-NN | 0.0 $_{\pm 0.0}$ | 0.0 $_{\pm 0.0}$ | 100.0 $_{\pm 0.0}$ |
| | | DSiRe | **0.0** $_{\pm 0.0}$ | **0.0** $_{\pm 0.0}$ | **100.0** $_{\pm 0.0}$ |
| Resnet-50 | [2, 100, 200, 1000] | Frobenius-NN | 15.86 $_{\pm 7.48}$ | 7.80 $_{\pm 3.50}$ | 96.70 $_{\pm 0.82}$ |
| | | DSiRe | **14.71** $_{\pm 5.66}$ | **5.9** $_{\pm 2.17}$ | **97.88** $_{\pm 0.82}$ |

pre-training weights, as Stable Diffusion 2.0 was *not* fine-tuned from a previous version. Tab. 3 shows that DSiRE performs well on Stable Diffusion 2.0, reaching around $80\%$ accuracy. This provides evidence for the correlation between the singular values and dataset size is not specific to one backbone alone.

**LLM.** To demonstrate that our method on modalities beside images, we experimented on GPT-2 (Radford et al., 2019) fine-tuned with LoRA on 50 micro-datasets derived from CNN-dailymail (Nallapati et al., 2016), each with 4 different sample sizes ([1, 50, 100, 500]). The results of our method in Tab. 3 show perfect accuracy, suggest that our method extends beyond images.

**ResNet.** To test desire on discriminative tasks, we experimnteed with ResNet-50 (He et al., 2016) finetuned using LoRA on 50 micro-datasets derived from CIFAR-100 (Krizhevsky et al., 2009), each with 5 different sample sizes ([2, 100, 200, 500, 1000]). Tab 3 presents the results, showing that DSiRE works in that case too.

# 6 ABLATION STUDIES

**Number of Micro-Datasets.** While our attack is data driven and requires access to the pre-trained model, we find that only a few examples are needed for DSiRe to perform well. E.g., in our medium data size range, our model can reach $86.4\%$ accuracy using only 5 micro-datasets for training. The full results, presented in Fig. 5, showcases that while more samples (fine-tuned models) improves the accuracy of our predictor, even a single micro-dataset is sufficient to achieve around $80\%$ accuracy. This shows that our method is robust to the number of micro-datasets used, even to very small numbers.

**DORA.** To evaluate robustness across more fine-tuning paradigms, we've included experiments with DORA (Liu et al., 2024) on ResNet-50 in the same settings as the ResNet-LoRA experiment. Tab. 7 shows DSiRe excels across different fine-tuning methods, including DORA.

**LoRA Rank.** We trained DSiRe for different LoRA ranks. Tab. 4 shows the results for medium and low data ranges. Our method is robust to the LoRA rank, achieving similar results in all 3 tested ranks for both ranges.

**Choice of classifier.** We tested different parametric and non-parametric classifiers, as shown in Fig. 4. In every case, we fit the classifier separately to each layer and select the predicted label via a majority vote rule of all layer-wise predictions. The only exception is the NN-full model uses a kNN classifier that fits all layers simultaneously. The results show that the choice of classifier affects the performance significantly. Furthermore, these results confirm our hypothesis from Sec. 3: while each layer is predictive of the dataset size, it is by combining all classifiers that we reach the best performance,

For more ablation studies on training steps, batch size, seeds, and learning rate, please refer to Appendix A.1.

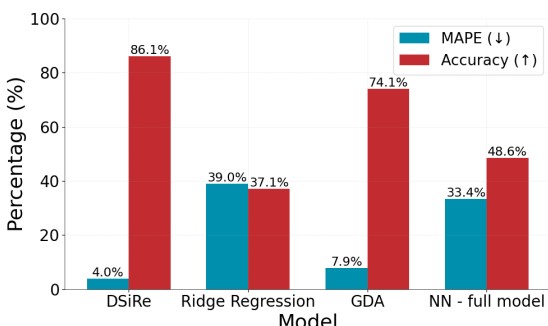

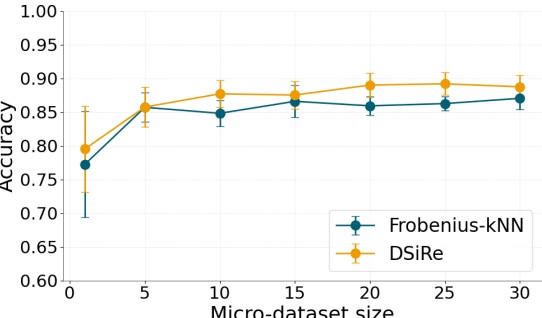

Figure 4: ***Performance of various predictors, dataset size range*** $(1-50)$***.*** DSiRe performs best by combining predictions from multiple layers, opposed to the NN - full model baseline, which uses the spectra of all layers together as features for a single prediction.

Figure 5: ***DSiRes Micro-Dataset Size vs. Accuracy, reported on medium data size range*** $(1-50)$***.*** Even a single micro-dataset is sufficient for DSiRe to reach $80\%$ accuracy. This demonstrates its effectiveness with limited training data.

## 7 Discussion

**Performance at Low Data Ranges.** While our approach shows promising results, there is room for improvement in lower data regimes, where DSiRe reaches less than $80\%$ accuracy. Improving these results will provide tighter upper bounds for membership inference and model inversion attacks.

**Data Driven Solution.** Our method is data driven as it requires training multiple models from each dataset size. However, our analysis shows there is correlation between the Frobenius norm and dataset size (see Fig. 2a). This insight could be a stepping stone in developing a data-free solution.

**Pre-training dataset size recovery.** Another interesting application of dataset size recovery is for pre-training cases. Lower bounding the required number of training set samples for foundation models will have a substantial impact on the research community. Answering this question would require scaling up our method to much larger dataset sizes and weight matrix dimensions.

## 8 Social impact

Research on our new task can positively impact both the research and digital arts communities. Establishing an upper bound for membership inference attacks can promote privacy aware deployment of fine-tuned models across different architectures and modalities. Determining the training dataset needed to train models with poor documentation can help inform researchers that need to collect expensive datasets for new fine-tuning tasks e.g., (Winter et al., 2024) and (Dai et al., 2023).

## 9 Conclusion

We introduced the novel task of dataset size recovery and proposed DSiRe, a method for learning a predictor for this task in fine-tuned models. Our extensive experiments demonstrate DSiRe's broad applicability across various modalities, network architectures, fine-tuning paradigms, and dataset sizes, including continuous ranges. We believe our work not only introduces a new capability but also provides valuable insights for research in model privacy and security, potentially serving as an upper bound for model inversion and membership inference attacks.

## 10 Reproducibility Statement

To ensure the reproducibility of our method and results, we provide detailed descriptions of the experimental results and implementation details in Section C.1. We have also included our code in the supplementary material. LoRA-WiSE benchmark will be made fully available upon acceptance.

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

# A   APPENDIX

## A.1   ADDITIONAL ABLATION STUDIES

### A.1.1   ROBUSTNESS TO LoRA HYPER-PARAMETERS

We provide more ablation studies of our method. Specifically, we test the training steps, batch size, learning rate and seeds used classifier type and used LoRA matrices.

**Batch Size.**  We ablate the batch size, results at shown in Tab. 5. Despite the change in batch size, DSiRe demonstrates robust performance, achieving a MAE score of 1.94 compared to the original 1.48. Additionally, the accuracy only decreases by less than 5%, indicating that our method maintains comparable effectiveness even with different batch sizes.

Table 4: ***DSiRe Performance with Different LoRA Ranks.***  Desire consistently achieves high accuracy across both low and medium ranges, indicating its robustness regardless of LoRA rank variations.

| Data Range | LoRA Rank | MAE ↓ | MAPE(%) ↓ | Acc(%) ↑ |
|---|---|---|---|---|
| | 8 | $0.43_{\pm 0.04}$ | $14.8_{\pm 2.3}$ | $66_{\pm 3.08}$ |
| $1-6$ | 16 | $0.42_{\pm 0.03}$ | $12.4_{\pm 1.11}$ | $67.7_{\pm 2.3}$ |
| | 32 | $0.36_{\pm 0.04}$ | $11.36_{\pm 1.55}$ | $69.30_{\pm 3.83}$ |
| | 16 | $1.67_{\pm 0.17}$ | $4.32_{\pm 0.46}$ | $84.04_{\pm 1.85}$ |
| $1-50$ | 32 | $1.48_{\pm 0.21}$ | $3.97_{\pm 0.73}$ | $86.10_{\pm 1.99}$ |
| | 64 | $1.41_{\pm 0.39}$ | $3.90_{\pm 1.30}$ | $86.58_{\pm 3.45}$ |

Table 5: DSiRe performance using different LoRA hyper-parameters. Medium data range

| Ablation | MAE↓ | MAPE(%) ↓ | Acc(%) ↑ |
|---|---|---|---|
| Batch size | $1.94_{\pm 0.26}$ | $9.35_{\pm 1.34}$ | $81.50_{\pm 2.55}$ |
| lr | $1.68_{\pm 0.23}$ | $5.15_{\pm 0.95}$ | $83.48_{\pm 2.14}$ |
| seeds | $1.52_{\pm 0.18}$ | $4.95_{\pm 0.85}$ | $82.48_{\pm 2.14}$ |
| Baseline | $1.48_{\pm 0.21}$ | $3.97_{\pm 0.73}$ | $86.10_{\pm 1.99}$ |

**Seeding.**  While in the standard recipe, all models use $seed = 0$, we also tested the case where all seeds were selected randomly. Tab. 5 shows that the variation in seeds only reduces accuracy by around 4%, and that MAE decreases by less than 0.5. This is not a small change, given that the gap between possible dataset size values is 10.

**Learning Rate.**  We ablate the learning rate, with results shown in Tab. 5. Despite the change in learning rate from our baseline, DSiRe demonstrates robust performance, achieving a MAE score of 1.68 compared to the original 1.48. Additionally, the accuracy only decreases by approximately 3%, indicating that our method maintains comparable effectiveness even with a different learning rate. These results further support DSiRe's resilience to variations in fine-tuning hyperparameters.

**Training Steps.**  To train DSiRe, we first fine-tune a set of LoRA models. These models follow a certain recipe, with a specific amount of training steps. To evaluate robustness, we tested DSiRe on models fine-tuned at different steps, with 1200 steps as our baseline. As shown in Tab 6, DSiRe consistintly achieves comparable

results across different fintuning steps. e.g. the MAE score ranges from $2.43$ at 300 steps to $1.40$ at 1400 steps, with accuracy variations within $10\%$.

Table 6: **DSiRe performance on different checkpoints of Stable Diffusion 1.5 rank 16 range** $1 - 50$

| #Steps | MAE↓ | MAPE(%)↓ | Acc(%)↑ |
|--------|------|----------|---------|
| 300 | 2.43 $\pm 0.20$ | 6.82 $\pm 0.78$ | 77.90 $\pm 1.49$ |
| 400 | 2.39 $\pm 0.20$ | 6.72 $\pm 0.76$ | 78.38 $\pm 1.49$ |
| 500 | 2.05 $\pm 0.15$ | 5.55 $\pm 0.59$ | 81.33 $\pm 1.60$ |
| 600 | 1.86 $\pm 0.10$ | 4.59 $\pm 0.34$ | 82.76 $\pm 0.86$ |
| 700 | 1.89 $\pm 0.21$ | 5.13 $\pm 0.77$ | 82.00 $\pm 2.01$ |
| 800 | 1.71 $\pm 0.29$ | 4.59 $\pm 0.89$ | 83.67 $\pm 2.68$ |
| 900 | 1.60 $\pm 0.22$ | 4.21 $\pm 0.69$ | 85.14 $\pm 2.04$ |
| 1000 | 1.62 $\pm 0.21$ | 4.69 $\pm 0.70$ | 85.10 $\pm 1.77$ |
| 1100 | 1.58 $\pm 0.19$ | 4.50 $\pm 0.90$ | 84.48 $\pm 1.32$ |
| 1200 | 1.48 $\pm 0.21$ | 3.97 $\pm 0.73$ | 86.10 $\pm 1.99$ |
| 1300 | 1.46 $\pm 0.15$ | 3.84 $\pm 0.51$ | 86.29 $\pm 1.55$ |
| 1400 | 1.40 $\pm 0.20$ | 3.73 $\pm 0.76$ | 86.76 $\pm 2.08$ |

# B  ADDITIONAL EXPERIMENTS AND AND FIGURES

**DORA.** To address concerns about robustness to different fine-tuning paradigms, we've included experiments with DORA (Differentiable Optimal Ranking Adaptation) (Liu et al., 2024) in addition to standard LoRA. We experimented with ResNet-50 using DORA in the same settings as the ResNet-LoRA experiment. Tab. 3 presents results that demonstrate DSiRe performs exceptionally well across various fine-tuning paradigms, including DORA. This provides further evidence of the method's versatility and robustness to different fine-tuning approaches.

Table 7: **Robustness of Dataset Size Recovery Methods on DoRA** DSiRe recovers dataset size more effectively than Frobenius-NN for the medium data range $(1 - 50)$ using Stable Diffusion 2.0. This supports the benefit from a more expressive representation given by

| Method | MAE↓ | MAPE(%) ↓ | Acc(%) ↑ |
|--------|------|-----------|----------|
| Frobenius-NN | 16.57 $\pm 13.3$ | 4.45 $\pm 2.29$ | 97.07 $\pm 1.69$ |
| DSiRe | 20.9 $\pm 12.04$ | 5.01 $\pm 1.95$ | 96.51 $\pm 1.62$ |

**Choice of LoRA matrices.** Seeing in Sec. 3 that not all layers are similar in behavior, we test to see if different LoRA matrices also capture different information. In Tab. 8, we find that indeed different LoRA matrices capture different information, and lead to substantially other performances. Unsurprisingly, we also find that using all the LoRA matrices combined yields the best result.

## B.1  ADDITIONAL FIGURES

To better understand the results on medium and higher data regimes we provide here the confusion matrices of DSiRe using $1 - 50$ and $1 - 1000$ training samples. We can see that most of the errors are in larger data classes.

Table 8: *DSiRe performance on different layers of LoRA of the UNet in Stable Diffusion 1.5, range* $1-50$:

| # Layer Type | $MAE \downarrow$ | $MAPE(\%) \downarrow$ | $Acc(\%)$ |
|---|---|---|---|
| A | $1.9_{\pm 0.29}$ | $5.63_{\pm 1.26}$ | $82.52_{\pm 2.20}$ |
| B | $1.57_{\pm 0.19}$ | $4.07_{\pm 0.65}$ | $84.90_{\pm 2.09}$ |
| BA | $1.61_{\pm 0.16}$ | $4.22_{\pm 0.47}$ | $85.00_{\pm 1.45}$ |
| full model | $1.48_{\pm 0.21}$ | $3.97_{\pm 0.73}$ | $86.10_{\pm 1.99}$ |

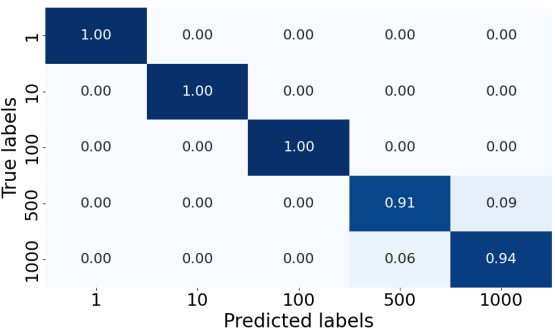

Figure 6: *DSiRe Confusion Matrix for Medium Data Range in a single experiment.* Illustrating DSiRes accuracy in the range of $1-50$ samples, shows that most of the errors are near misses, highlighting DSiRe's precision in dataset size recovery.

Figure 7: **DSiRe Confusion matrix in High data regime.** Illustrating DSiRe's accuracy in the range data size $(1-1000)$ for a single experiment, showing that most predictions are correct or near misses, highlighting the DSiRe's precision in dataset size recovery.

## C  LoRA-WiSE Benchmark

We present the LoRA Weight Size Evaluation (LoRA-WiSE) benchmark, a comprehensive benchmark specifically designed to evaluate LoRA dataset size recovery methods, for generative models. More specifically, it features the weights of 2350 Stable Diffusion (Rombach et al., 2022) models, which were LoRA fine-tuned by a standard, popular protocol (Ruiz et al., 2023; dre). Our benchmark includes versions 1.5 and 2 of Stable Diffusion, having 2050 and 300 trained models for each version respectively.

We fine-tune the models using three different ranges of dataset size: (i) Low data range: $1-6$ images. (ii) Medium data range: $1-50$ images. (iii) High data range: $1-1000$. For each range, we use a discrete set of fine-tuning dataset sizes. In the low and medium ranges, we also provide other versions of these benchmarks with different LoRA ranks and backbones. See Tab.9 for the precise benchmark details.

For our low data range set, we choose Concept101 (Kumari et al., 2023), a previously collected set of micro-datasets ($3-15$ images) designed for personalization research. For our medium and high data ranges we use different classes of ImageNet (Deng et al., 2009) as the data source. This selection of datasets aims to ensure that the fine-tuned models are drawn from a diverse set of concepts, spanning various categories.

Each micro-dataset is used to fine-tune the models for each dataset size. The images are randomly selected from the micro-dataset. Each Stable Diffusion model consists of 132 adapted layers (pairs of $A_i, B_i$), including various layer types, such as self-attention, cross-attention, and MLPs. We save $A_i, B_i$ separately, i.e., each model provides a total of 264 unique weight matrices. We then split each range of this new benchmark (low, medium, and high ranges) into train and test sets based on the micro-datasets. for more details see appendix C.1.

Table 9: ***LoRA WiSE Benchmark Overview.*** The dataset comprises over 25,000 weights checkpoints drawn from more than 2000 independent LoRA models, spanning different dataset sizes, backbones, ranks, and personalization sets.

| Data Range | Dataset Sizes | Source | Backbone | LoRA Rank | # of Models |
|---|---|---|---|---|---|
| Low | $1, 2, 3, 4, 5, 6$ | Concept101 | SD 1.5 | 8
16
32 | 300
300
300 |
| Medium | $1, 10, 20, 30, 40, 50$ | ImageNet | SD 1.5 | 16
32
64 | 300
300
300 |
| | | | SD 2 | 32 | 300 |
| High | $1, 50, 100, 500, 1000$ | ImageNet | SD 1.5 | 32 | 250 |

## C.1 IMPLEMENTATIONS DETAILS

**LoRA-WiSE.** we now elaborate the implementations details of the LoRA-WiSE bench dataset. As the Pre-Ft models we use `runwayml/stable-diffusion-v1-5` and `stabilityai/stable-diffusion-2` (Rombach et al., 2022). We fine-tune the models using the PEFT library (Mangrulkar et al., 2022). We use the script `train_dreambooth_lora.py` (Ruiz et al., 2023) with the diffusers library (von Platen et al., 2022).

For each regime, we have 50 micro-datasets with varying sizes. For each size $s$ in the regime's dataset sizes' range, we sample $s$ images from each micro-dataset, and train SD 1.5 on this resulted dataset

we use the standard recipe to fine-tune the models in all ranges(dre) see tabs. 10 and 11. we use batch size 8 for a range of 1-1000 for computational resources and 1000 training steps. in the ablations, we don't change any hyper-parameter except the ablate one.

Each model took approximately 30-50 minutes to fine-tune. We used GPUs with 16-21GB of RAM, such as the NVIDIA RTX A5000. The DSiRe process, however, does not require GPUs and can run on CPUs.

**Full Fine-Tuning.** For the pre-fine-tuned models, we use `runwayml/stable-diffusion-v1-5`. We employ the script `train_text_to_image.py` for training.

we use the standard recipe to fine-tune the models in the range (dre), we choose 50 random classes from ImageNet to fine-tune the models on medium regime ($\{1, 10, 20, 30, 40, 50\}$).

**LLM** We fine-tune the GPT-2 model from Hugging Face on 50 different datasets derived from CNN-DailyMail (Nallapati et al., 2016), using varying dataset sizes [1, 50, 100, 500]. LoRA is applied with a rank of 16 and an alpha of 32, and the model is trained for 100 steps.

**ResNet** We fine-tune ResNet50 from the torchvision models on CIFAR-100. LoRA is used with a rank of 32, alpha of 32, and targets the conv1, conv2, and conv3 layers. CIFAR-100 is split into 50 distinct datasets, with each dataset composed of two combined classes. The ResNet is fine-tuned on these datasets across various sizes ([2, 100, 200, 500, 1000]) for 250 steps.

**Experiment Settings.** In addition to the experiment settings described in Section 5, we used the following configurations for our models:

Table 10: *ranges 1-6 and 1-50*

| Name | Value |
|---|---|
| `lora_rank` ($r$) | $r$ |
| `lr` | $1e-4$ |
| `batch_size` | 1 |
| `gradient_accumulation_steps` | 1 |
| `learning_rate_scheduler` | Constant |
| `training_steps` | 1400 |
| `warmup_ratio` | 0 |
| `dataset` | `imagenet`(Deng et al., 2009) `concept101`(Kumari et al., 2023) |
| `seeds` | 0 |

Table 11: *range 1-1000 Hyper-parameters*

| Name | Value |
|---|---|
| `lora_rank` ($r$) | 32 |
| `lr` | $1e-4$ |
| `batch_size` | 8 |
| `gradient_accumulation_steps` | 1 |
| `learning_rate_scheduler` | Constant |
| `training_steps` | 1000 |
| `warmup_ratio` | 0 |
| `dataset` | `imagenet` |
| `seeds` | 0 |

- For models in the ranges 1-6 and 1-50, we used the checkpoint at iteration 1200.

- For models in the range 1-1000, we used the checkpoint at iteration 1000.

-We used a fixed seed of 42 to split the train and test data for every experiment.

**Layer weigh matrices** In line with our analysis see Sec.3, given $n$ example weights for each $A_i, B_i$ we wish to build a separate classifier for each one. Knowing the relation between the singular values and the dataset size, we decompose each matrix using the singular value decomposition (SVD), and use the ordered set of singular values as features for our classifiers. Formally, we note the singular values of $A_{ij}$ as $\Sigma_{A_{ij}}$ and the singular values of $B_{ij}$ as $\Sigma_{B_{ij}}$. We include the singular values of $B_{ij} \cdot A_{ij}$ denoted as $\Sigma_{B_{ij} \cdot A_{ij}}$. Additionally, our observations indicate that the product $B_i \cdot A_i$ also provides useful information for data size recovery. Thus, for each LoRA matrix, we obtain a dataset with $n$ samples, where each sample is a vector of singular values $\Sigma_{ij}$, paired with a corresponding label $y_j$. Our method then trains three separate kNN-classifiers with $K = 1$ for each layer over (i) $A_i$ (ii) $B_i$ and (iii) $B_i A_i$. At inference time, the predictions from all classifiers are merged by majority voting.

