# OpenReview forum: "Dataset Size Recovery from Fine-Tuned Weights"
_ICLR.cc/2025/Conference — ICLR 2025 Conference Withdrawn Submission_

### Official Review · Reviewer_4DiE · 2024-11-04

**Soundness:** 3
**Presentation:** 3
**Contribution:** 2
**Rating:** 3
**Confidence:** 4

**Summary:**

The paper proposes the problem of dataset size recovery for fine-tuned foundation models and consequently a strategy to infer dataset size using spectral analysis of the weight matrix. A benchmark is designed to evaluate various approaches for this problem and their proposed method is evaluated on it.

**Strengths:**

1. The problem of dataset size recovery for foundation models is interesting.
2. The correlation of dataset size to the Frobenius norm and singular values of the weight matrices is relevant.
3. A benchmark with pre-trained weight matrices of foundation models for dataset recovery is released.

**Weaknesses:**

1. The analysis of the correlation between dataset size and the Frobenius norm and the singular values is underwhelming. It is not clear if this trend holds across different model architectures, and if so, no theoretical evidence is advanced for this correlation.
2. The proposed method for dataset size recovery is way too simple to offer any insights.
3. The authors only study dataset size recovery for foundation models fine-tuned with a few samples. However, this problem is very general and should be explored in a broader framework.

**Questions:**

Please refer to weaknesses above.

---

### Official Review · Reviewer_8NQV · 2024-11-04

**Soundness:** 2
**Presentation:** 3
**Contribution:** 2
**Rating:** 5
**Confidence:** 3

**Summary:**

This paper investigates the challenge of estimating the training data size of a fine-tuned pre-trained model. The authors find that the norms and spectral properties of model weight are correlated with the dataset size used during fine-tuning. Based on this insight, they propose an algorithm called DSiRe. DSiRe utilizes a nearest-neighbours approach to classify each layer independently, with the final dataset size prediction determined by a majority vote across layers.

**Strengths:**

This work studies an interesting topic, which aims to find out the training data size from a given fine-tuned model.

The proposed DSiRe method shows promising results in predicting dataset sizes, suggesting that the spectral and norm-based characteristics of fine-tuned weights are indeed useful signals for this task.

This work offers a practical resource for future research by proposing a benchmark kit.

**Weaknesses:**

The authors formulate dataset size recovery as a classification problem, whereas it may be more appropriate to approach this as a regression problem. Since dataset size is inherently a continuous variable, a regression framework might offer a more precise and interpretable estimation than classification.

The number of samples (1~1000) used in the experiment is very limited, which may not get reliable conclusions in real-world scenarios.

The study does not discuss the potential effects of data augmentation on dataset size recovery. Given that data augmentation is a common practice in model training, understanding its impact on the proposed method's accuracy is crucial. It would be valuable to include experiments or discussions on how data augmentation could alter spectral and norm properties in fine-tuned weights.

While the paper explores estimating dataset size, it would be insightful to discuss how this information could impact model inversion techniques or the general machine learning community. For example, does knowing the dataset size improve an adversary's ability to reconstruct original training samples?

**Questions:**

Please kindly see the weakness section

---

### Official Review · Reviewer_DCUx · 2024-11-04

**Soundness:** 2
**Presentation:** 3
**Contribution:** 3
**Rating:** 5
**Confidence:** 4

**Summary:**

The paper proposes a new task, called "dataset size recovery," which aims to identify the size of the fine-tuned dataset based on changes in model weights before and after fine-tuning. The authors define a data-driven pipeline to achieve this: several fine-tuned weights and their corresponding dataset sizes are provided as training samples, and during testing, a newly fine-tuned model is given. The goal is to predict the dataset size of this test model. Specifically, they propose extracting spectral features from the model weights and using these features to predict dataset size with a nearest neighbor algorithm. For experiments, the authors introduce a new benchmark named LoRA-WiSE, where various stable diffusion models are fine-tuned with LoRA parameterizations across different dataset sizes. They demonstrate the efficacy of the proposed algorithm by presenting mean absolute error (MAE) scores across three data regimes: low (up to 6 samples), medium (up to 50 samples), and high (up to 1,000 samples).

**Strengths:**

1. The paper proposes an interesting and, to the best of my knowledge, novel problem: recovering the dataset size based on fine-tuned model weights. This approach seems potentially useful for tasks such as model inversion and membership inference attacks.

2. The paper constructs a large-scale dataset, including 2,000 diverse LoRA fine-tuned models along with corresponding fine-tuning dataset information, which could be valuable for future research.

3. The observed correlation between fine-tuning dataset size and both the weight norm and spectrum provides meaningful insights. The results presented with the proposed method appear reasonable across the benchmark.

**Weaknesses:**

1. The method appears to predict the fine-tuning dataset size for a given model only when “training samples”—pairs of model weights and corresponding fine-tuning dataset sizes—are available. However, it remains unclear how one would construct these training samples in practice, particularly without prior information about the actual fine-tuning dataset used by the model.

2. Beyond dataset size, other factors likely influence the norms and spectra of the learned weights, such as the diversity of the fine-tuning dataset or its divergence from the pretraining dataset. Without direct knowledge of the fine-tuning data, these factors remain uncontrolled. For instance, a model fine-tuned on a large but homogeneous dataset may exhibit more overfitting than one fine-tuned on a small yet diverse dataset, resulting in higher norms or spectral values. This raises concerns regarding the method’s practical applicability.

3. As shown in Figure 2, the distinctions between different fine-tuning dataset sizes diminish as dataset size increases, making it unclear how effective the method remains for larger datasets.

4. The experiments focus solely on a stable diffusion model, leaving questions about the method’s generalizability to other model types. Additionally, why is the method restricted to fine-tuned weights? Could it be extended to estimate the dataset size for a model trained from scratch, and would the trends observed in Figure 2 apply in that context?

**Questions:**

Please see the weaknesses part.

---

### Official Review · Reviewer_hb6b · 2024-11-06

**Soundness:** 2
**Presentation:** 3
**Contribution:** 3
**Rating:** 3
**Confidence:** 4

**Summary:**

This paper introduces a new task of dataset size recovery, which aims to infer the size of the training dataset used to fine-tune a pre-trained model.
Through experiments, the authors uncover a clear negative correlation between dataset size and the norm and spectrum of the fine-tuning weight matrices.
Leveraging this insight, they propose the DSiRe algorithm to predict dataset size based on these spectral features.
Additionally, the authors propose the LoRA-WiSE benchmark for evaluating on this task.

**Strengths:**

* This paper introduce a novel task correlating to model inversion and membership inference attacks. The size of the training dataset will produce extra knowledge for these tasks. Besides, the authors propose a benchmark for evaluation.
* The paper is well-written and easy to follow.
* The authors provide code for reproducibility check.

**Weaknesses:**

1. **Lack of theoretical support:** Although the authors reveal a quantitative relationship between dataset size and the characteristics of fine-tuning weight matrices, their evaluation is limited to diffusion tasks, lacking broader empirical evidence. Furthermore, the authors do not provide theoretical insights or justification to explain why this relationship exists.
2. **Experiments:** The authors should validate the effectiveness of the proposed method across a wider range of tasks, such as image classification.
3. **Experiments:** The authors claim that knowing the size of dataset could aid in model inversion and membership inference attacks. Could the authors provide additional experiments to support this claim?

**Questions:**

My questions are listed in Weaknesses section.

---

### Note · Authors · 2024-11-17

**Comment:**

We sincerely thank the reviewers for their detailed and insightful feedback on our paper. After reflecting on the overall scores, we have decided to withdraw the submission. We truly value the time and effort the reviewers invested in evaluating our work.

**Withdrawal Confirmation:**

I have read and agree with the venue's withdrawal policy on behalf of myself and my co-authors.